# Enhanced Diabetic Wound Healing Using Electrospun Biocompatible PLGA-Based Saxagliptin Fibrous Membranes

**DOI:** 10.3390/nano12213740

**Published:** 2022-10-25

**Authors:** Chen-Hung Lee, Shu-Chun Huang, Kuo-Chun Hung, Chia-Jung Cho, Shih-Jung Liu

**Affiliations:** 1Division of Cardiology, Department of Internal Medicine, Chang Gung Memorial Hospital-Linkou, Chang Gung University College of Medicine, Taoyuan 33305, Taiwan; 2Department of Physical Medicine and Rehabilitation, New Taipei Municipal Tucheng Hospital, New Taipei City 23652, Taiwan; 3Department of Physical Medicine & Rehabilitation, Chang Gung Memorial Hospital, Taoyuan 33305, Taiwan; 4College of Medicine, Chang Gung University, Kwei-Shan, Taoyuan 33302, Taiwan; 5Institute of Biotechnology and Chemical Engineering, I-Shou University, Kaohsiung 84001, Taiwan; 6Department of Orthopedic Surgery, Bone and Joint Research Center, Chang Gung Memorial Hospital-Linkou, Taoyuan 33305, Taiwan; 7Department of Mechanical Engineering, Chang Gung University, Taoyuan 33302, Taiwan

**Keywords:** PLGA-based fibrous membranes, electrospinning, saxagliptin, release characteristics, diabetic wound repair

## Abstract

Delayed diabetic wound healing is an adverse event that frequently leads to limb disability or loss. A novel and promising vehicle for the treatment of diabetic wounds is required for clinical purposes. The biocompatible and resorbable poly (lactic-co-glycolic acid) (PLGA)-based fibrous membranes prepared by electrospinning that provide a sustained discharge of saxagliptin for diabetic wound healing were fabricated. The concentration of released saxagliptin in Dulbecco’s phosphate-buffered saline was analyzed for 30 days using high-performance liquid chromatography. The effectiveness of the eluted saxagliptin was identified using an endothelial progenitor cell migration assay in vitro and a diabetic wound healing in vivo. Greater hydrophilicity and water storage were shown in the saxagliptin-incorporated PLGA membranes than in the pristine PLGA membranes (both *p* < 0.001). For diabetic wound healing, the saxagliptin membranes accelerated the wound closure rate, the dermal thickness, and the heme oxygenase-1 level over the follicle areas compared to those in the pristine PLGA group at two weeks post-treatment. The saxagliptin group also had remarkably higher expressions of insulin-like growth factor I expression and transforming growth factor-β1 than the control group (*p* = 0.009 and *p* < 0.001, respectively) in diabetic wounds after treatment. The electrospun PLGA-based saxagliptin membranes exhibited excellent biomechanical and biological features that enhanced diabetic wound closure and increased the antioxidant activity, cellular granulation, and functionality.

## 1. Introduction

Delayed wound healing is a main secondary adverse event in the case of diabetes that frequently leads to limb disability and loss [1]. Impairments of the healing of diabetic wounds that contribute to the amputation of a lower limb are among the most common poor outcomes that are related to diabetes mellitus and lead to a mortality that is even higher than those associated with many cancers [2]. The hindered healing of a diabetic wound is also the main cause of various complications, such as the impeded functioning of keratinocytes and fibroblasts due to hyperglycemia and disrupted epithelialization and wound closure [3,4]. A novel and promising scheme for the treatment of diabetic wounds is clinically required.

Saxagliptin, a dipeptidyl-peptidase inhibitor (DPP-4 inhibitor) that is used for the management of diabetes, prevents the degradation of the endogenous secreted glucose-dependent insulinotropic peptide and glucagon-like peptide-1 that are associated with insulin release following intake. In addition to blocking the degradation of peptides, DPP-4 inhibitors provide many other physiological substrates, such as neurohormones, chemokines, and cytokines, that can improve metabolism, vascular function, and pleotropic performance [5,6,7]. Additionally, DPP4 inhibitors remarkably speed wound closure and improve ulcer-related poor outcomes that are associated with cellulitis, osteomyelitis, and infection [8,9,10]. However, the systemic administration of saxagliptin may cause adverse complications [11,12], including runny or stuffy nose, cough, headache, and stomach pain [13,14]. This evidence motivated the adoption of a saxagliptin-loaded dressing for the management of diabetic wounds.

Poly (lactic-co-glycolic acid) (PLGA) is a resorbable biomaterial that has great potential for use as a scaffold in tissue engineering and as therapeutic delivery vehicles [15,16]. The salient features of PLGA that have attracted particular attention include its excellent biodegradability/biocompatibility, the feasibility of its use in offering sustained drug release and protecting drugs from degradation, the surface modifiability to enable effective biological interactions between numerous types of drugs, and its usefulness in targeting specific cells or organs [17,18,19].

We developed PLGA-based saxagliptin-incorporated fibrous membranes using the electrospinning technique and hypothesized that the topical treatments with the saxagliptin-eluting membranes could accelerate diabetic wound closure and re-epithelialization. The spun fibrous membranes were evaluated for their morphology, hydrophilicity, tensile strength, and water storage capacity. The release characteristics and therapeutic activity of saxagliptin from the membranes were investigated using high-performance liquid chromatography (HPLC) and an endothelial progenitor cell (EPC) migration assay in vitro. The remedial efficacy of the fibrous saxagliptin membranes in diabetic wound dressing was also evaluated on an in vivo rat model.

## 2. Materials and Methods

### 2.1. Fabrication of Electrospun Fibrous Membranes

The polymer employed was PLGA (Resomer RG 503, Boehringer, Germany) with a lactide:glycolide ratio of 50:50 and an intrinsic viscosity of 0.4. Saxagliptin hydrate (C_18_H_25_N_3_O_2_·H_2_O) and hexafluoro-2-propanol (HFIP) were also used (Sigma-Aldrich, Saint Louis, MO, USA).

Two fabricated membranes, saxagliptin-based PLGA (PLGA, 0.240 g; saxagliptin, 0.040 g) and pristine PLGA electrospun fibrous membranes (PLGA, 0.280 g), were both dissolved in 1000 µL of HFIP. The mixtures were then electrospun at room temperature by a lab-made spinning device, which included a high-voltage power supply, a syringe with a pump, and a grounded collection plate. In line with the optimum process parameters from our previous studies, the following conditions were employed: voltage supply: 35 kV; flow rate: 60 µL per min; distance between the syringe and the collector: 15 cm) [20,21]. All electrospun fibrous PLGA membranes were stored in a vacuum oven (Model CKN-30, Cheng-Huei Co., Taipei, Taiwan) at 40 °C for 72 h to vaporize the HFIP solvents.

### 2.2. Scanning Electron Microscopic (SEM) Observation

The sizes of the fibrous distribution and pore area were obtained by analyzing SEM (Jeol JSM–7500F, Tokyo, Japan) images of at least 50 randomly chosen areas in each test sample (n = 3) [20] using Image J image software (National Institutes of Health, Bethesda, MD, USA).

### 2.3. Mechanical Properties of Electrospun Fibrous Membranes

A Lloyd tensiometer (AMETEK, Berwyn, PA, USA) was used to evaluate the mechanical properties, including the tensile strength (MPa, breaking force (N)/cross-sectional area of sample (mm^2^)) and elongation at breakage (%, increase in length at breaking point (mm)/initial length (mm) × 100) of the fabricated fibrous products.

### 2.4. Wetting Angles

The wetting angles of both the saxagliptin/PLGA and pristine PLGA membranes were obtained using a water contact angle analyzer (First Ten Angstroms, Portsmouth, VA, USA) that was equipped with a video monitor.

### 2.5. Water Uptake Ability of Saxagliptin/PLGA and Pristine PLGA Electrospun Fibers

The water uptake ability of both electrospun fibers over 0.5, 1, 2, 3, 8, and 24 h was obtained using the water content formula.

### 2.6. In Vitro Release of Saxagliptin

An HPLC assay (Hitachi L-2200 Multisolvent Delivery System, Tokyo, Japan) was utilized to determine the release characteristics of saxagliptin from the electrospun fibers. The process was conducted using an XBridge C_18_ 5 μm, 4.6 × 150 mm HPLC column (Waters, Milford, MA, USA) for saxagliptin separation.

### 2.7. Endothelial Progenitor Cells (EPCs) Migration Assay

Transwell filters (Costar, Cambridge, MA, USA) with 8.0 µm pores were used in the migration assay, as described elsewhere [22,23]. The EPCs were obtained from the Laboratory of Molecular Pharmacology (Chang-Gung University, Taoyuan, Taiwan) and were placed on top of the transwell filter. Solutions containing different days of elute from the saxagliptin/PLGA membrane or DPBS were placed under the cell permeation filter. Following an incubation period of 12 h, the cells that migrated through the filter were stained and counted. Data were acquired from five randomly chosen areas of the eluents from the saxagliptin/PLGA membranes at each time point.

### 2.8. Evaluation of Diabetic Wound Healing

All animal-related procedures were performed with the approval of the Institutional Animal Experiment Committees of Chang Gung University (IACUC Approval No. CGU14-045; 06/10/2014 approved). Diabetes was induced in 14 Sprague–Dawley rats using streptozotocin (STZ) (Sigma, St Louis, MO, USA). One week after STZ injection, the animals were confirmed to have hyperglycemia (≥300 mg/dl) prior to the assessment of diabetic wound healing. Wounds were then created on the back of each rat.

Seven animals were treated with PLGA-based saxagliptin membranes for diabetic wound healing, and another seven were treated with pristine PLGA electrospun fibrous membranes as the control group.

### 2.9. Immunofluorescence and Western Blot Analysis

For immunostaining, the samples were washed in PBST and blocked with 2% bovine serum albumin for 30 min at room temperature. The samples were then incubated overnight at 4 ℃ with primary antibodies against heme oxygenase-1(HO-1) (1/500, sc-136960, Santa Cruz Biotechnology, Santa Cruz, CA, USA). They were subsequently stained with fluorescently labeled AF 546 goat anti-mouse secondary antibodies (1/500, Life Technologies, Carlsbad, CA, USA) overnight at 4 ℃. Then, 4,6-diamidino-2-phenylindole (DAPI) (nuclear stain, 1/2000 dilution in PBST, 2 h) was added the next day. The specimens were washed and mounted for visualization on a Leica confocal microscope. The mean fluorescence intensity (MFI) was obtained as a mean over the regions of interest (ROIs) using ImageJ software. The MFI of the target proteins was normalized to the MFI of the DAPI as an internal control to account for sample-to-sample variations.

For western blotting, a fixed amount of protein (30 µg) in dodecyl sulfate polyacrylamide gel electrophoresis (SDS-PAGE) buffer was sonicated and subjected to electrophoresis on 12% SDS-polyacrylamide gels. After transfer to the polyvinylidene difluoride (PVDF) membranes, the proteins were incubated with primary antibodies (insulin-like growth factor 1(IGF-1) receptor, 1/1000, #9750, Cell Signaling Technology, Danvers, MA, USA; anti-transforming growth factor (TGF)-β1, 1/2000, ab179695, Abcam, Cambridge, UK). The membranes were rinsed and incubated with anti-rabbit peroxidase-conjugated IgG secondary antibodies (#711-035-152, Jackson ImmunoResearch, West Grove, PA, USA) that were diluted to 1/10,000 in tris-buffered saline (TBST) (0.1% Tween 20) for one hour. Densitometric analysis of the protein expression was normalized to the expression of the loading control glyceraldehyde-3-phosphate dehydrogenase (GAPDH) (1/10,000, ab8245, Abcam) using ImageJ software. The statistical analyses were based on a minimum of three repetitions.

### 2.10. Statistics and Data Analysis

All data are shown as mean ± standard deviation. The means of the continuous variables and normally distributed data were compared by the unpaired Student’s t test; otherwise, the Mann–Whitney U test was performed. Differences were determined to be statistically significant at *p* < 0.05. The data were analyzed using SPSS software (version 17.0 for Windows; SPSS Inc., Chicago, IL, USA).

## 3. Results and Discussion

### 3.1. Electrospinning of the Electrospun Fibrous Membranes

Figure 1A,B reveal the SEM photographs of the electrospun fibrous membranes at 3000× magnification. The spun diameters of the saxagliptin/PLGA fibers (444.5 ± 153.2 nm) were considerably smaller than those of the pristine PLGA fibers (997.7 ± 351.8 nm) (*p* < 0.001). In addition, the pore area of the electrospun saxagliptin PLGA fibrous membranes was significantly lower than that of the pristine PLGA membranes (3560 ± 1745 × 10^3^ nm^2^ and 8843 ± 3605 × 10^3^ nm^2^, for the PLGA-based saxagliptin membranes and pristine PLGA membranes, respectively) (*p* < 0.001) (Figure 1C–F).

The results of the mechanical properties shown in Figure 2A suggest that both groups exhibited comparable tensile strengths (3.13 ± 0.10 MPa for saxagliptin/PLGA vs. 3.14 ± 0.11 MPa for the PLGA group) (*p* = 0.950). Additionally, the pristine electrospun fibers showed higher elongation at breakage (218.6 ± 24.2%) than the saxagliptin/PLGA electrospun fibers (134.6 ± 6.7%) (*p* = 0.004). The concentration of PLGA, applied voltage, and flow rate are parameters that critically affect the size of the electrospun fiber diameter and the uniformity [24]. With a fixed high voltage (35 kV) and flow rate (3.6 mL per hour) [21], the decrease in the PLGA concentration in the solution yielded fibers of smaller diameter, mainly owing to the reductions in the polymer viscosity and the degree of chain entanglements [25,26,27]. Larger fibers have significantly higher ultimate tensile strength, independent of the strain rate [28]. Additionally, the tensile moduli of the electrospun PLGA-based saxagliptin and pristine PLGA fibers were similar to that of rat skin (approximately 3.2 MPa) [29]. The spun fibrous scaffold was able to provide sufficient support for tissue regeneration during the healing process. Figure 2B shows the percentage of the water content in the saxagliptin/PLGA and pristine PLGA fibers following DPBS immersion at different time points. The pristine fibrous membranes had their highest water content (around 55%) between half and one hour, while the saxagliptin/PLGA fibers retained a high percentage (>138%) of water content for the first 24 h, reaching their highest water content (193 ± 14%) in the 24th hour. The capacity of the saxagliptin/PLGA fibers for storing water always exceeded that of the pristine PLGA fibers (all *p* < 0.001) (See Appendix A for the water contents in both groups).

Figure 3 shows the water contact angles of the resorbable PLGA membranes with (Figure 3A) and without saxagliptin (Figure 3B). The measured water contact angles for the saxagliptin/PLGA and the pristine fibrous membranes were 56.48 ± 2.88° and 116.91 ± 2.94°, respectively. Mixing with saxagliptin significantly enhanced the hydrophilic property of the pristine PLGA fibrous membranes (Figure 3C) (*p* < 0.001). 

PLGA is regarded as a hydrophobic compound, requiring the utilization of organic solvents for product development. It has been commonly used to encapsulate both water-soluble and water-insoluble pharmaceuticals for the purpose of drug delivery [30,31]. The presence of saxagliptin, a relatively hydrophilic compound with water solubility (around 18 mg/mL) [32], in electrospun PLGA fibrous membranes can increase their water uptake capacity and hydrophilicity. The hydrophilic feature also favors the speed and quality of wound healing, reducing eschar formation [33], maintaining cellular interaction [34], and enhancing cell migration over the wound surface [35]. Furthermore, the PLGA-based saxagliptin membranes exhibited suitable flexibility and extensibility, demonstrating their tunable utilization in wound dressings that allow skin contraction during the healing process. 

### 3.2. In Vitro Release Curves of the Saxagliptin and the EPC Migration Assay

The in vitro daily release curves of saxagliptin are plotted in Figure 4A. The drug-loaded PLGA membranes released saxagliptin continuously for 30 days, with an initial release burst on the first day (245.2± µg/mL), followed by a second peak period from the third week (86.5 ± 2.7 µg/mL) and a steady drug concentration (>23.4 µg/mL) until day 30. More cells migrated on the eluents of day 1 (419.7 ± 12.1 cells/mm^2^) (*p* = 0.001), day 7 (361.1 ± 24.1 cells/mm^2^) (*p* = 0.007), and day 14 (347.3 ± 16.3 cells/mm^2^) (*p* = 0.004) than those treated with DPBS (262.4 ± 23.7 cells/mm^2^), according to the EPC transwell migration assay (Figure 4B–F). Drug release kinetics are influenced by the drug content and properties of the polymeric carrier, as the surface segregation of the drug precedes the initial burst release [36,37,38]. The rapid release profile arises mainly from the simple diffusion of loaded drugs as a result of a lack of interactions between the PLGA and blended medications and the highly porous structure of the electrospun fibrous membrane [39]. Administered saxagliptin would be able to enhance the endothelium-dependent relaxation, prevent endothelial apoptosis, and reduce endothelial impairment [40,41].

### 3.3. Wound Healing and Histological Examination

Figure 5A–F display a representative image of the healed diabetic wound on an animal from each group (PLGA-based saxagliptin and pristine PLGA fibrous membranes) on different days (days zero, three, and fourteen) post-treatment. The wound areas were comparable in the two groups on the first day (82.9 ± 5.0 vs. 85.2 ± 3.8 mm^2^, *p* = 0.566). Wound areas in the pristine PLGA membranes were larger than those in the saxagliptin/PLGA group on days three and fourteen (Day 3: 64.1 ± 5.0 vs. 43.3 ± 3.0 mm^2^, *p* = 0.003; Day 14: 33.3 ± 1.3 vs. 19.6 ± 1.0 mm^2^, *p* < 0.001) **(**Figure 5G). Wound healing is a combination of dynamic processes, which are regulated by many growth factors and the secretion of cytokines from cells [42,43]. DPP-4 has been found to affect peri-wound inflammation, re-epithelialization, extracellular matrix secretion, and skin fibrosis and is a potential target for promoting wound healing [44]. It has many physiological components, such as chemokines, cytokines, and neurohormones that are associated with metabolism and vascular function [5,45]. By impeding the enzymatic degradation of the above components, DPP-4 inhibitors can induce pleiotropic effects [46]. Therefore, dressing a diabetic wound with saxagliptin-eluting fibrous membranes increased the rate of its healing over that in the control group.

The histological images show that the saxagliptin/PLGA fibrous membranes (Figure 6A) increased cell proliferation over that in the pristine PLGA fibers group (Figure 6B). Dressing with a saxagliptin/PLGA scaffold for two weeks yielded a significantly higher epidermal thickness in the histological sections of the saxagliptin group, because of the cell infiltration (indicated by the asterisk) between the epidermis and the dermis (82.0 ± 3.8 µm vs. 16.7 ± 3.0 µm, *p* < 0.001) (Figure 6C). These results were confirmed by the MFI associated with immunostaining for protein expression (HO-1: Figure 6D,E; Autofluorescence: Figure 6H,I; DAPI nuclear stain in Figure 6F,G). The expression over the follicle areas (white asterisk) (Figure 6D,E) in the saxagliptin/PLGA group (0.88 ± 0.19) (Figure 6J,K) was significantly higher than in the pristine PLGA group (0.51 ± 0.10) (*p* = 0.005, Figure 6L). Wound healing is a response to injury and involves many steps of overlapping events during tissue injury [46]. The wound repair process consists of numerous steps, including hemostasis/coagulation, inflammatory cell recruitment, the proliferative phase, and the maturation phase [47,48]. After hemostasis and the inflammatory phase (around two to five days), the proliferative phase dominates the covering of the wound surface (re-epithelialization), which requires the proliferation and migration of keratinocytes [49,50]. The regenerative process is highly regulated by the expression of several proteins and chemoattractants, including the recruitment of sufficient cells and the activation of regenerative pathways [51]. DPP4 is expressed in keratinocytes, subsets of hematopoietic progenitor cells, and endothelial cells and macrophages [52]. The dermis and epidermis are also the main constituents of the skin, where the major sources of enzymes are dermal fibroblasts and epidermal keratinocytes. The activity of DPP4 varies among wound repair phases. High DPP4 levels in diabetic wounds sustain a persistent inflammatory status that impairs wound healing [53]. The results herein suggest that saxagliptin released from drug-loaded membranes enhances wound healing by inducing the speed of fibroblasts and keratinocytes to the diabetic wound area. In addition, diabetes reduces skin renewal capacity as a result of epidermal dysfunction and premature skin aging [54]. When skin is damaged or injured in diabetic models, the hair follicle stem cells are deactivated, causing migration of the cells to the wound site to be slow and their differentiation into epidermal cells inadequate, delaying re-epithelialization of the wound [55,56,57,58]. Oxidative stress caused by hyperglycemia is considered to be the most important mechanism of the long-term complications of diabetes, as well as an obstacle to stem-cell proliferation [59,60]. Heme oxygenae-1 (HO-1) attenuates oxidative stress by catalyzing the oxidative degradation of heme to form free iron biliverdin, and carbon monoxide [61,62]. The overexpression of HO-1 is protective because of its potential antioxidant, anti-inflammatory, and antiapoptotic activity [63]. The antiapoptotic effect of saxagliptin on inflammatory cytokines ameliorates the influence of oxidative load and contributes positively to a host’s antioxidant defenses [64,65]. The adoption of electrospun saxagliptin/PLGA fibrous membranes therefore ameliorated oxidative stress and promoted healing in diabetic wounds.

The saxagliptin/PLGA group yielded remarkably higher protein expression levels of insulin-like growth factor 1 (IGF-1) (intensity ratio: 0.98 ± 0.06 vs. 0.56 ± 0.02 *p* = 0.009) (Figure 7A,B) and transforming growth factor (TGF)-β1 (intensity ratio: 0.73 ± 0.05 vs. 0.50 ± 0.07 *p* < 0.001) (Figure 7C,D) in wounds after treatment. During wound healing, patients with diabetes lack the activity of the increased IGF-1expression that contributes to the cellular granulation process [66,67]. TGF-β1 is involved in several cellular functions, including the control of cell growth, cell differentiation, cell proliferation, and apoptosis [68,69]. A lower level of TGF-β1 indicates a poorer prognosis in patients with diabetic foot ulcers [70,71]. This work demonstrated that the release of the DPP4 inhibitor from the spun fibrous membranes may be responsible for the increase in the IGF-1 and TGF-β1 levels over those in the control group during skin wound healing [72,73].

## 4. Conclusions

Biocompatible and resorbable saxagliptin-incorporated PLGA fibrous membranes that sustainably released saxagliptin for 30 days were fabricated by electrospinning. The effective and sustained release of saxagliptin enhanced the migration of endothelial progenitor cells and accelerated diabetic wound healing. The electrospun saxagliptin/PLGA fibrous membranes had higher hydrophilicity and water storage capacity than the pristine PLGA membranes. The saxagliptin membranes sped up the wound closure rate and increased the dermal thickness and heme oxygenase-1 level over follicle areas as compared to the pristine PLGA group. The saxagliptin group also showed much higher expressions of HO-1, IGF-1, and TGF-β1 in the diabetic wounds than did the control group. The saxagliptin/PLGA fibrous membranes exhibited biomechanical and biological features that promoted diabetic wound closure and increased antioxidant activity, cellular granulation, and functionality. Additional preclinical studies will be needed to verify the potential clinical relevance of this current work.

## Figures and Tables

**Figure 1 nanomaterials-12-03740-f001:**
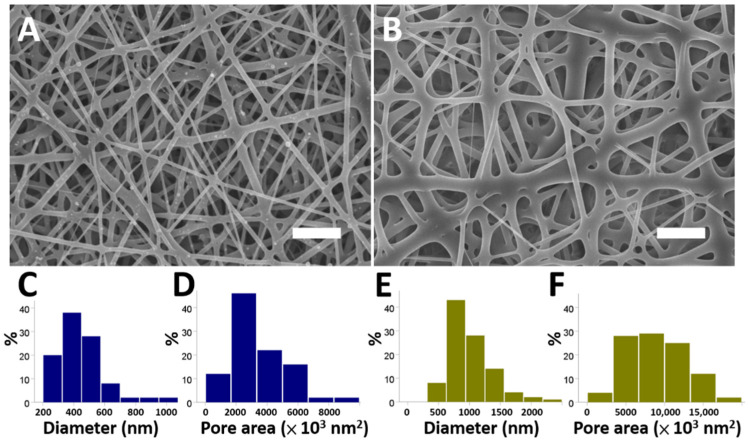
The electrospun fibers were evaluated using SEM. The morphology of the saxagliptin/PLGA fibers (**A**), and pristine PLGA fibers (**B**). The diameters and pore areas of the electrospun PLGA with saxagliptin (**C**,**D**) or pristine PLGA membranes (**E**,**F**) were assessed using Image J software (Scale bar: 5 μm).

**Figure 2 nanomaterials-12-03740-f002:**
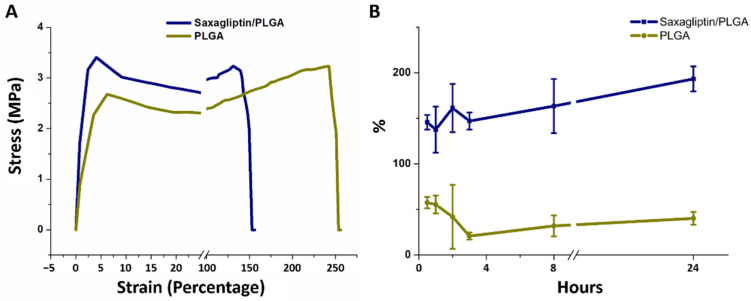
Stress–strain curves of the saxagliptin/PLGA and pristine PLGA resorbable electrospun fibrous membranes (**A**). Percentage water content of the fibers over 24 h (**B**).

**Figure 3 nanomaterials-12-03740-f003:**
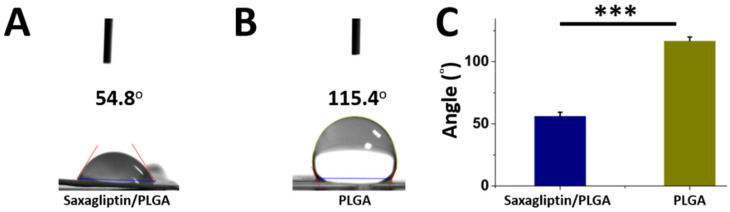
Measured water contact angles. (**A**) PLGA-based saxagliptin and (**B**) pristine PLGA resorbable fibrous membranes. Measured contact angle of both groups (**C**). *** *p* < 0.001.

**Figure 4 nanomaterials-12-03740-f004:**
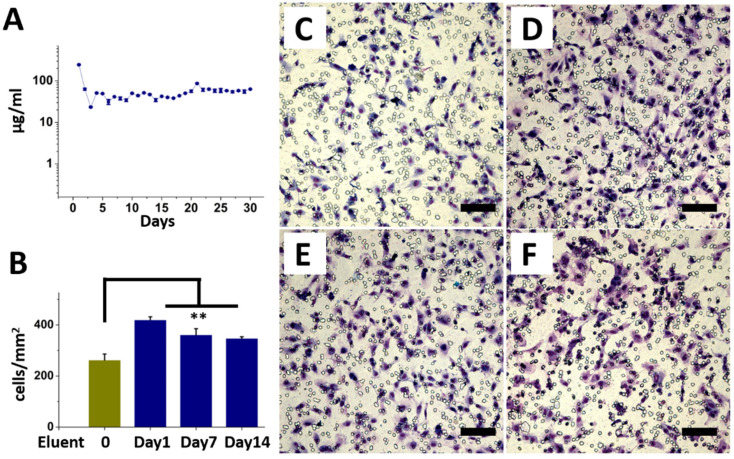
In vitro release of saxagliptin (**A**). Eluents from the PLGA-based saxagliptin electrospun fibrous membranes for the EPC invasion assay (**B**). EPCs treated with DPBS only (**C**), with eluents on day 1 (**D**), day 7 (**E**), and day 14 (**F**). (Scale bar = 100 µm). ** *p* < 0.01.

**Figure 5 nanomaterials-12-03740-f005:**
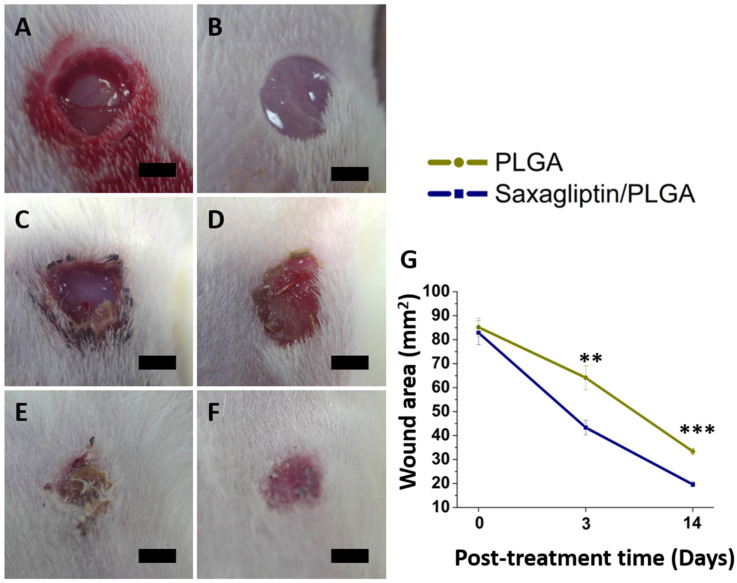
Progress of the healing wound after treatment with saxagliptin/PLGA and pristine PLGA group on day 0 (**A**,**B**), day 3 (**C**,**D**), and day 14 (**E**,**F**). Calculated wound areas on different days (**G**). (Scale bar = 5 mm). (Scale bar = 5 mm). ** *p* < 0.01, *** *p* < 0.001.

**Figure 6 nanomaterials-12-03740-f006:**
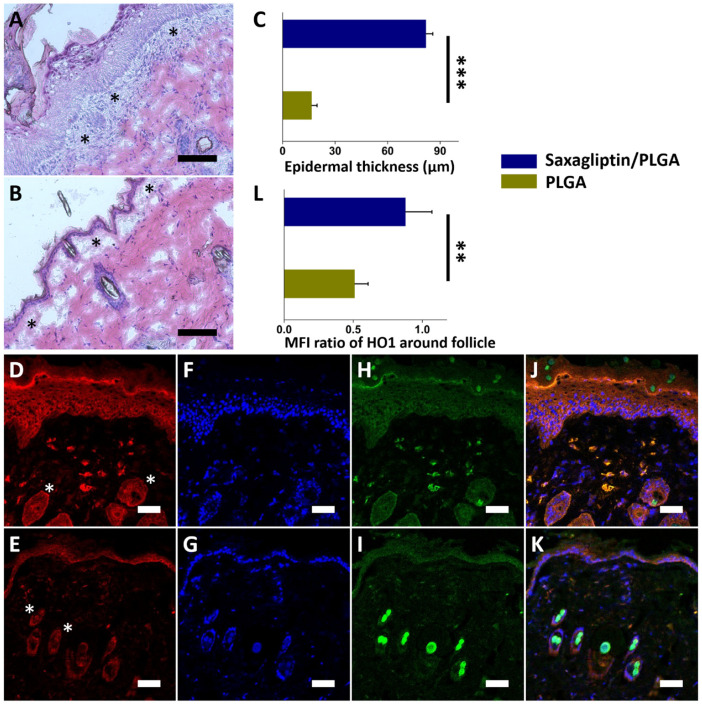
Histological pictures of the PLGA-based saxagliptin group (**A**) and pristine PLGA (**B**) on day 14. Histological sections of the treated wounds in both groups revealed a significantly higher depth of epidermis and cell infiltration (black asterisks) in the saxagliptin/PLGA group (**C**). Immunofluorescence of HO-1 in the saxagliptin/PLGA (**D**) and pristine PLGA (**E**) groups. Images of the DAPI-labeled nuclei (blue) (**F**,**G**), autofluorescence (**H**,**I**), and merged images for both groups (**J**,**K**). HO1 immunocytochemistry was quantified relative to that of DAPI using the mean fluorescence intensity (MFI) (**L**) over the area of follicles (white asterisks) (Scale bar = 100 μm). ** *p* < 0.01, *** *p* < 0.001.

**Figure 7 nanomaterials-12-03740-f007:**
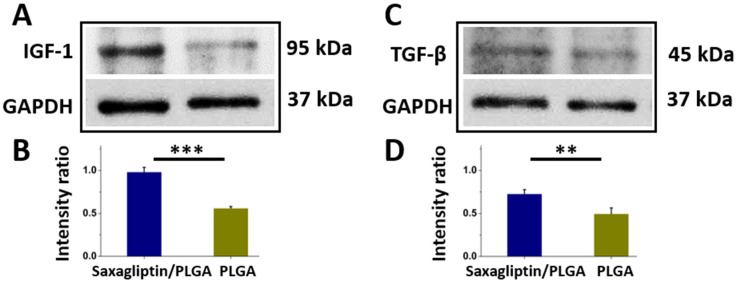
Western blotting and the calculated ratio of IGF-1(**A**,**B**) and TGF-β1(**C**,**D**) from the histologic section on day 14. The intensity ratio was obtained by densitometry as the ratio of the target protein to GAPDH density. ** *p* < 0.01, *** *p* < 0.001.

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
