# Peer review of "Enhanced Diabetic Wound Healing Using Electrospun Biocompatible PLGA-Based Saxagliptin Fibrous Membranes"

_nanomaterials, 2022, doi:10.3390/nano12213740_

Round 1

Reviewer 1 Report

Manuscript titled “Enhanced Diabetic Wound Healing Using Nanofibrous Biocompatible PLGA-based Saxagliptin Membranes” seems interesting, covering an area of concern for researchers working in the field of wound care. Authors performed not only in-vitro, but also in-vivo study to evaluate the membrane’s potential for suitability as wound dressing. However, manuscript needs sufficient amendments to be readable and understandable by the readers. My few suggestions are as follows;

1.    Introduction part is well written, however the literature cited is outdated (2017 or earlier). More recent work related to wound care, electrospinning, and in-vitro/in-vivo studies should be included. Moreover the language should be simple, taking care of grammar and sentence completion.

Authors can search recent literature by applying filter, for example recent 2 or 3 years. For their convenience, they can follow some of the following articles;

https://doi.org/10.1038/s41598-019-49132-x; https://doi.org/10.1021/acsanm.0c01562.

2.    First sentence of materials section is incomplete. Such mistakes should be corrected throughout the text.

3.    Fibers with diameter in the range of 1000 nm are not nanofibers. Authors can write electrospun fibers instead of electrospun nanofibers. In title, authors also claim nanofibers, I would suggest not to use term nanofibers. Title should also be revised.

Author Response

Q1. Introduction part is well written, however the literature cited is outdated (2017 or earlier). More recent work related to wound care, electrospinning, and in-vitro/in-vivo studies should be included. Moreover, the language should be simple, taking care of grammar and sentence completion. Authors can search recent literature by applying filter, for example recent 2 or 3 years. For their convenience, they can follow some of the following articleshttps://doi.org/10.1038/s41598-019-49132-x; https://doi.org/10.1021/acsanm.0c01562.

Answer: The articles published in recent years have been cited and addressed. Poly (lactic-co-glycolic acid) (PLGA) is a resorbable biomaterial that has great potential for use as a scaffold in tissue engineering and as therapeutic delivery vehicles15-16.

  1. Kharaghani D, Gitigard P, Ohtani H, Kim KO, Ullah S, Saito Y, Khan MQ and Kim IS. Design and characterization of dual drug delivery based on in-situ assembled PVA/PAN core-shell nanofibers for wound dressing application. Scientific Reports. 2019;9:1-11. (doi.org/10.1038/s41598-019-49132-x)
  2. Ullah S, Ullah A, Lee J, Jeong Y, Hashmi M, Zhu C, Joo KI, Cha HJ and Kim IS. Reusability comparison of melt-blown vs nanofiber face mask filters for use in the coronavirus pandemic. ACS Applied Nano Materials. 2020;3:7231-7241. (doi.org/10.1021/acsanm.0c01562)

Q2. First sentence of materials section is incomplete. Such mistakes should be corrected throughout the text.

Answer: The first sentence of materials section has been revised. We have copied the text again here for convenience: “The polymer used was PLGA (Resomer RG 503, Boehringer, Germany) with a lactide:glycolide ratio of 50:50 and an intrinsic viscosity of 0.4. Saxagliptin hydrate (C18H25N3O2 ·H2O) and hexafluoro-2-propanol (HFIP) were also used (Sigma-Aldrich, Saint Louis, U.S.A.).”

Q3. Fibers with diameter in the range of 1000 nm are not nanofibers. Authors can write electrospun fibers instead of electrospun nanofibers. In title, authors also claim nanofibers, I would suggest not to use term nanofibers. Title should also be revised.

Answer: The title and whole manuscript has been revised as electrospun fibers instead of electrospun nanofibers.” Enhanced Diabetic Wound Healing Using Electrospun Biocompatible PLGA-based Saxagliptin Fibrous Membranes.”

Reviewer 2 Report

The authors have investigated a membrane consisting of PLGA and saxagliptin to be used as a wound dressing or diabetic ulcers.

The authors have combined material steps with in vitro and in vivo studies.

General opinion

The manuscript should be improved to allow a better understanding of the methods, certain sentences need to be rephrased.

Detailed opinion

Line 65: "topical treatment...." the sentence should be in plural

Line 82: the type of the used equipment need to be described where applicable (oven, electrospinning)

Line 109: The references 27,28 does not contain the migration assay description, the method needs to be described in detail, or even using a graphic abstract as the method is confusing. 

Line 117, 118: please rephrase

Line 180 - 182: please rephrase

Line 188: "at" should be instead of "in"

Line 215: "the" is missing

Line 225: "administering"

Line 236,237 please rephrase

Line 273: What is also the main constituent?

Line 290: Defense is a verb, defence is the noun 

Line 323 "speeded up"

Author Response

The manuscript should be improved to allow a better understanding of the methods, certain sentences need to be rephrased.

Q1. Line 65: "topical treatment...." the sentence should be in plural

Answer: The sentence has been revised. Topical treatments with electrospun PLGA-based saxagliptin fibrous membranes have been hypothesized to accelerate diabetic wound closure and re-epithelialization.

Q2. Line 82: the type of the used equipment need to be described where applicable (oven, electrospinning)

Answer: The type of the used equipment has been provided as follows: “Two fabricated membranes, saxagliptin-based PLGA (PLGA, 0.240 g; saxagliptin, 0.040 g) and pristine PLGA electrospun fibrous membranes (PLGA, 0.280 g), were both dissolved the substances in 1000 µL of HFIP. The mixtures were then electrospun at room temperature by a lab-made spinning device, which included a high-voltage power supply, a syringe with a pump, and a grounded collection plate. In line with the optimum process parameters from our previous studies, the following condition was employed: voltage supply: 35kV; flow rate: 60 µL per min; distance between the syringe and the collector: 15 cm)20 21. All electrospun nanofibrous PLGA membranes were stored in a vacuum oven (Model CKN-30, Cheng-Huei Co., Taipei, Taiwan) at 40oC for 72 hours to vaporize the HFIP solvents.”

  1. Lee C-H, Liu K-S, Chang S-H, Chen W-J, Hung K-C, Liu S-J, Pang J-HS, Juang J-H, Chou C-C and Chang P-C. Promoting diabetic wound therapy using biodegradable rhPDGF-loaded nanofibrous membranes: CONSORT-compliant article. Medicine. 2015;94.
  2. Lee C-H, Hsieh M-J, Liu S-C, Chen J-K, Liu S-J, Hsieh I-C, Wen M-S and Hung K-C. Novel bifurcation stents coated with bioabsorbable nanofibers with extended and controlled release of rosuvastatin and paclitaxel. Materials Science and Engineering: C. 2018;88:61-69.

Q3. Line 109: The references 27,28 does not contain the migration assay description, the method needs to be described in detail, or even using a graphic abstract as the method is confusing. 

Answer: The migration assay description has been provided in detail as following sentences. “The EPCs were placed on top of the transwell filter, and solutions containing different days of elute from saxagliptin/PLGA membrane or DPBS were placed under the cell permeation filter. Following an incubation period of 12 hours, the cells that have migrated through the filter were stained and counted. Data were acquired from five randomly chosen areas of the eluents from saxagliptin/PLGA membranes at each time point.”

Q4. Line 117, 118: please rephrase

Answer: The sentences have been revised as “The diabetic wound healing process involving 14 Sprague-Dawley rats was induced using streptozotocin (STZ) (Sigma, St Louis, MO, USA). One week after STZ injection, animals were confirmed hyperglycemia (≥ 300 mg/dl) prior to assessment of diabetic wound healing.”

Q5. Line 180 - 182: please rephrase

Answer: The sentence has been rephrased as “The electrospun fibrous scaffold was able to provide sufficient support for tissue regeneration during the healing process.”

Q6. Line 188: "at" should be instead of "in"

Answer: The sentence has been rephrased as “The capacity of the saxagliptin/PLGA nanofibers for storing water exceeded that of pristine PLGA fibers at all times (all p < 0.001) (See Supplemental Table S1 for the water contents in both groups).”

Q7. Line 215: "the" is missing

Answer: The word has been corrected. “The saxagliptin/PLGA membranes released saxagliptin continuously for 30 days, with an initial burst release on the first day (245.2± µg/ml), followed by a second peak period from the third week (86.5 ± 2.7 µg/ml), after the released concentration (> 23.4 µg/ml) until day 30.”

Q8. Line 225: "administering"

Answer: The word has been corrected as the following sentence. “Administering saxagliptin can enhance endothelium-dependent relaxation, prevent endothelial apoptosis, and reduce endothelial impairment40 41.”

  1. Wu C, Hu S, Wang N and Tian J. Dipeptidyl peptidase‑4 inhibitor sitagliptin prevents high glucose‑induced apoptosis via activation of AMP‑activated protein kinase in endothelial cells. Molecular Medicine Reports. 2017;15:4346-4351.
  2. Wang H, Zhou Y, Guo Z, Dong Y, Xu J, Huang H, Liu H and Wang W. Sitagliptin attenuates endothelial dysfunction of Zucker diabetic fatty rats: implication of the antiperoxynitrite and autophagy. Journal of cardiovascular pharmacology and therapeutics. 2018;23:66-78.

Q9. Line 236,237 please rephrase

Answer: The Line 236 and 237 have been revised and we have copied the text again here for convenience: “The wound areas were found comparable in the two groups on the first day (82.9 ± 5.0 vs. 85.2 ± 3.8mm2, p = 0.566).”

Q10. Line 273: What is also the main constituent?

Answer: The main constituent of skin has been provided. “The dermis and epidermis are also main constituents of the skin, where the major sources of enzymes are dermal fibroblasts and epidermal keratinocytes.”

Q11. Line 290: Defense is a verb, defence is the noun 

Answer: The word has been revised to “defences”. “The anti-apoptotic effect of saxagliptin on inflammatory cytokines ameliorates the influence of oxidative load and contributes positively to a host’s antioxidant defences64, 65.”

  1. Helal MG, Zaki MMAF and Said E. Nephroprotective effect of saxagliptin against gentamicin-induced nephrotoxicity, emphasis on anti-oxidant, anti-inflammatory and anti-apoptic effects. Life sciences. 2018;208:64-71.
  2. Chen Y-T, Tsai T-H, Yang C-C, Sun C-K, Chang L-T, Chen H-H, Chang C-L, Sung P-H, Zhen Y-Y and Leu S. Exendin-4 and sitagliptin protect kidney from ischemia-reperfusion injury through suppressing oxidative stress and inflammatory reaction. Journal of translational medicine. 2013;11:1-19.

Q12. Line 323 "speeded up"

Answer: The sentence has been revised as the following. “Saxagliptin membranes speeded up wound closure rate and increased dermal thickness and heme oxygenase-1 level over follicle areas above the values exhibited by the pristine PLGA group.”

Reviewer 3 Report

The paper is interesting - however, many aspeects of wound healing have been not addressed. Plenty of papers in teh field. However, I understand that the scope of your work is limited. Good work!

Author Response

The paper is interesting - however, many aspects of wound healing have been not addressed. Plenty of papers in the field. However, I understand that the scope of your work is limited. Good work!

Answer: Thank you for this excellent suggestion. We agree that future work will need to confirm the relevance of these observations using clinical studies. We respectfully suggest that clinical in vivo studies are outside the scope of the current work, which focused on understanding the mechanistic links between saxagliptin/PLGA substrate and the increase of IGF-1 and TGF-β1 levels over those in the control group during skin wound healing. To make this important point to the reader, we have added the following text: “Future work will require identification of the validation of these observations using more in vivo pre-clinical studies to verify potential clinical relevance.”

Round 2

Reviewer 2 Report

The manuscript can be accepted in the present form.

Author Response

Thank you very much for the comment. The English of the manuscript has been thoroughly checked again to improve the readability.